# Implementation of Water Tracers in the Met Office Unified Model

Alison J. McLaren<sup>1</sup>, Louise C. Sime<sup>1</sup>, Simon Wilson<sup>2</sup>, Jeff Ridley<sup>3</sup>, Qinggang Gao<sup>1,4</sup>, Merve Gorguner<sup>5</sup>, Giorgia Line<sup>3</sup>, Martin Werner<sup>6</sup> and Paul Valdes<sup>5</sup>

Correspondence to: Alison McLaren (aliren@bas.ac.uk)

#### Abstract.

10

There is an increasing need to understand how water is cycled and transported within the atmosphere to aid water management. Here, atmospheric water tracers are added to the Met Office Unified Model (UM) to allow tracking of water within the model. This requires the implementation of water tracers in the following parts of the model code: large-scale advection, surface evaporation, boundary layer mixing, large-scale precipitation (microphysics), large-scale clouds, stochastic physics and convection. A single water tracer is found to track all water in the model to a high degree of accuracy during a 35-year simulation; the differences are typically less than 10<sup>-16</sup> kg kg<sup>-1</sup> at the end of every timestep, prior to a very small adjustment to prevent the build up of numerical error. The increase in computing time for each water tracer is between 3.1 and 3.8% depending on the model resolution. The model development is tested by using the water tracers to find the sources of precipitation in a historical UM simulation. As expected, the majority of precipitation is found to be sourced directly from the ocean, with the recycling of water over land becoming increasingly important downwind across continents. The UM results for the mean evaporative source properties of precipitation are comparable to those of the ECHAM6 atmospheric model, with some interesting local differences over Antarctica, Greenland and the Indian monsoon region. Finally, global model hydrological fluxes are derived from the water tracers to show the amount of precipitation sourced from the land and ocean separately, which illustrates the additional information that can be provided from the new development.

#### 1 Introduction

A major part of the global hydrological cycle consists of water evaporating at the surface, being transported through the atmosphere, and eventually returning to the surface as rain or snow. The water vapour in the atmosphere has a mean residence

<sup>&</sup>lt;sup>1</sup>Ice Dynamics and Palaeoclimate, British Antarctic Survey, Cambridge, U.K.

<sup>&</sup>lt;sup>5</sup> National Centre for Atmospheric Science, Computational Modelling Service, University of Reading, U.K.

<sup>&</sup>lt;sup>3</sup>Met Office Hadley Centre, Exeter, U.K.

<sup>&</sup>lt;sup>4</sup>Now at School of Geography, Earth and Atmospheric Sciences, University of Melbourne, Australia

<sup>&</sup>lt;sup>5</sup>School of Geographical Sciences, University of Bristol, U.K.

<sup>&</sup>lt;sup>6</sup>Alfred Wegener Institute, Helmholtz Centre for Polar and Marine Research, Bremerhaven, Germany

time of 8-10 days (Gimeno et al., 2021) and during this time it can be advected thousands of kilometres (e.g. Fiorella et al., 2021). Thus, the moisture source and sink locations for precipitation can differ greatly. For water management purposes, it is becoming increasingly important to understand how water moves through the atmosphere between regions, at global and regional scales, and how it may change in the future (Rockström et al., 2023). This is because humans are affecting the water cycle both directly, via processes such as extraction and irrigation, and indirectly through climate change and land surface changes (Abbott et al., 2019; Allan et al., 2020; Dorigo et al. 2021).

Gimeno et al. (2012) reviewed the various methods for tracking atmospheric water pathways, which they divided into three groups: analytical/box models, numerical water vapour tracers, and physical water vapour tracers (i.e. water isotopes). The numerical tracer approach has been used in 'offline', using an Eulerian or Lagrangian framework, and 'online' models (Dominguez et al., 2020). 'Offline' systems use water tracking models forced by output from General Circulation Models (GCMs) or reanalyses (e.g. Tuinenburg et al., 2020; van der Ent et al., 2010, 2014). 'Online' models involve embedding numerical water tracers (hereafter termed 'water tracers') into a GCM and is the focus of this paper. The online method allows a greater complexity in the representation of the processes that affect water (Dominguez et al., 2020), plus it has the advantage that the tracers respond to their forcing at every model timestep rather than using time-averaged offline fields. Including online water tracers in climate model simulations also allows predicted changes in water transport to be easily investigated. However, the online method comes with the added computational cost of running a GCM, although recent water tracer developments have made the method more efficient as discussed below. Traditional online water tracers that track evaporation from prescribed regions are also unable to provide detailed spatial information on individual source regions for specific events.

40

Water tracers track moisture around a GCM and are impacted by the same physical processes that affect water in the model. However, unlike the model's prognostic water fields, they are passive tracers in that they do not affect the physics or dynamics of the model. Water tracers, including water isotopes, have been successfully added to many GCMs (recent examples include Nusbaumer et al., 2017, and Cauquoin et al., 2019) following the early work of Joussaume et al. (1984) and Koster et al. (1986). Stable water isotopes can be viewed as a special type of water tracer that undergo fractionation during phase changes. Typically, non-isotopic water tracers have been used to track the source region of precipitation by tagging water evaporating from prescribed regions. This technique has been used to investigate the sources of Antarctic and Greenland precipitation to aid the interpretation of ice core isotopic measurements (e.g. Werner et al., 2001; Delaygue et al., 2000; Noone and Simmonds, 2002). Outside of the polar regions, water tracers have been used to investigate precipitation source regions including over the Indian Monsoon region (Tharammal et al., 2023), the Eurasian continent (Numaguti, 1999), North America and India (Bosilovich and Schubert, 2002) and during an atmospheric river over North America (Nusbaumer and Noone, 2018). Water tracers have also been included in high resolution regional models and used to investigate sources of specific precipitation events, such as work by Insua-Costa and Miguez-Macho (2018) and Winschall et al. (2014); the latter reference also includes

a direct comparison with a Lagrangian method. Following the nomenclature of Gao et al. (2024; hereafter 'G24'), the type of water tracer used in the above studies is termed here as a 'prescribed-region' tracer.

Recently, Fiorella et al. (2021) introduced 'process-orientated' water tracers which included a water tracer that captures the precipitation weighted mean value of an evaporative property (e.g. latitude or sea surface temperature, SST). G24 further developed this method to focus on ocean-sourced precipitation and to ensure water tracer conservation; they used the scheme to investigate ocean source properties of Antarctic precipitation in the ECHAM6 model. Following G24, these tracers are referred to here as 'scaled-flux' tracers to distinguish them from the prescribed-region tracers. A comparison between prescribed-region and scaled-flux tracers is presented in G24. In summary, the scaled-flux tracer produces a more precise estimate of the mass weighted mean evaporative property in a highly efficient way compared with the prescribed-region method. Therefore, using these online tracers to find the mean location of evaporative sources of precipitation has become less computationally expensive. The disadvantage of the scaled-flux method is that it does not capture any information about the variability of the source property. However, analysing mean values over short timescales (e.g. daily data) does provide some information on the variability.

This paper describes the implementation of non-isotopic water tracers in the Met Office Unified Model (Brown et al., 2012). The UM is an operational atmospheric model used across a range of timescales from weather to climate; it is the atmospheric component of the UK Earth System Model (UKESM; Sellar et al., 2019). The UM water tracer development is the first stage of a larger project to add water isotopes to the UKESM. As noted by Noone and Sturm (2010), implementing water tracers in a GCM comprises most of the effort required for adding isotopes to a GCM. Water isotopes were added to an earlier version of the UM as part of the work by Tindall et al. (2009) to include water isotopes in the coupled model HadCM3 (Pope et al., 2000; Gordon et al., 2000); HadCM3, which is relatively computationally efficient to run, continues to be well used by the palaeoclimate community (Valdes et al, 2017; Oger et al, 2023). However, the isotope code written for HadCM3 was never permanently included in the UM, and is therefore, out of date with more recent UM versions. Hence, in order to add water tracers and isotopes to the state-of-the-art model version, the UM code development described here is brand new. A key priority of the work is to ensure water tracers (and eventually water isotopes) become a permanent option in the UM to ensure the longevity of this development.

The purpose of this paper is to document the UM water tracer implementation and to provide evidence that the scheme is working as expected. For a general description of how water tracers and isotopes are added to a GCM, the reader is referred to Noone and Sturm (2010). The paper is structured as follows. The water tracer implementation is described in section 2, together with an assessment of the computing cost of the water tracers. Definitions of the different types of water tracers are also provided in Section 2, plus details of the simulations used to test the development. Section 3 contains the results from the

test simulations including a comparison with the ECHAM6 atmospheric model. The paper ends with conclusions and an outlook for future work in section 4.

#### 100 2 Model Description





The UM solves the non-hydrostatic, fully compressible deep-atmosphere equations of motion using a semi-implicit semi-Lagrangian method (Wood et al., 2014). The model uses a regular longitude-latitude grid with terrain-following hybrid height coordinates. The 'Global Atmosphere' (GA) 7.0/7.1 science configuration of the UM is documented in Walters et al. (2019). The most recent scientific configuration (GAL9.0, Willett et al., 2025b) is used in this study. The UM is run with the Joint UK Land Environment Simulator (JULES) land surface model (Best et al., 2011; Clark et al., 2011). However, in this study, water tracers are not included in JULES and remain in the atmosphere component only.

# 2.1 Basic Water Tracer Code Development

Water tracers are added to the UM such that they evolve according to the same processes that act on the model's prognostic water fields, with the aim of following the model's water as precisely as possible. This requires adding water tracers to the following schemes: large-scale advection, surface evaporation, boundary layer mixing, large-scale precipitation (microphysics), large-scale clouds, stochastic physics and convection. Unlike water, the water tracers are passive and do not impact on the model physics or dynamics. The water tracers are held in an array with the number of water tracers set by the user. The first water tracer in the array mimics the prognostic water in the model and is named here as the 'normal water tracer'. This tracer is used to continually test the water tracer simulation as it can be compared directly to the model's water fields. The other water tracers in the array can then be used to trace specific water around the model or water properties (e.g. prescribed-region or scaled-flux tracers), and, in the future, to model water isotopes.

In the UM configuration used here, water is modelled using four prognostic fields for vapour, liquid condensate, ice condensate and large-scale rain (see Walters et al., 2019, for details); note, snow remains purely a diagnostic quantity. Equivalent prognostic fields for water tracers are defined as the product of the water field and the ratio of the water tracer to water. For example, the water tracer specific humidity equivalent  $(q_{wt})$  is

$$q_{wt} = R_q \ q \tag{1}$$

where  $R_q$  is the ratio of water tracer vapour to water vapour and q is the specific humidity. Water tracer fields for the other three water phases are similarly defined; therefore, adding m water tracers to the model requires an additional 4\*m tracer fields.

When water changes phase in the model, the water tracers are updated using the water tracer to water ratio of the source phase. For condensation as an example, the water tracers are updated as






$$q_{wt}^{n+1} = q_{wt}^n - R_q \Delta q \tag{2}$$

$$qcl_{wt}^{n+1} = qcl_{wt}^n + R_a \Delta q (3)$$

where  $\Delta q$  is the change in q due to condensation,  $qcl_{wt}$  is the water tracer specific liquid condensate. n and n+1 indicate the values before and after the phase change respectively.

The large-scale advection of water tracers uses the same methods as prognostic water. In the model configuration used here, the semi-Lagrangian interpolation to the departure points uses a bi-cubic interpolation in the horizontal and a cubic Hermite interpolation in the vertical (Walters et al., 2019). Mass conservation of moist prognostics is enforced globally using the Optimised Conservative Filter (OCF) scheme of Zerroukat and Allen (2015) and the same scheme is used for the normal water tracer. Vertical transport of the water tracers in the convection and microphysics scheme is calculated as the relevant water tracer to water ratio multiplied by the water flux.

The water tracer code generally involves replicating the model's prognostic water code for the array of water tracers. For example, the water tracers pass through the entire convection and microphysics schemes following the vertical transport of water and undergoing the same phase changes. The water tracers are mostly updated in separate new subroutines to avoid overcomplicating the base model code. This means that the order of calculations for water tracers can be slightly different to water which due to numerical error will cause tiny differences between the normal water tracer and water. At the end of a timestep (20 mins), the difference is typically less than  $10^{-16}$  kg kg<sup>-1</sup> when using double precision. Although this is an insignificant amount, this can grow quickly over time and hence a very small adjustment is made to the water tracers at the end of each timestep so that the normal water tracer exactly matches the equivalent water field. To ensure that this adjustment does not hide any issues, the code also contains a check to ensure that this adjustment remains less than  $10^{-10}$  kg kg<sup>-1</sup> over a timestep. In the 35-year long simulation that is used in this paper, the adjustment remains less than this level of  $10^{-10}$  kg kg<sup>-1</sup> at every timestep showing that the water tracer code is correctly tracing the water in the model to a high degree of accuracy.

The water tracer code was included in UM release 13.2 in March 2023 following the code passing the Met Office code review process (Met Office Simulations Systems Working Practices, 2024). At this stage, normal water tracers were available to be used in runs using the GA 8.0 scientific configuration (Willett et al., 2025a). Since then, the code has been further developed in branches to: a) work with the Global Atmosphere Land 9.0 (GAL9.0) configuration (Willett et al., 2025b), which will be the basis for the scientific configuration used in the simulations provided to the Coupled Model Intercomparison Project 7; and b) set up the prescribed-region and scaled-flux types of water tracers as described in section 2.3.

#### 2.2 Computational Cost of Water Tracers




To assess the increase in computing cost caused by adding water tracers, month-long simulations with varying numbers of normal water tracers were run on the Met Office Cray XC40 supercomputer. Simulations were carried out at two resolutions: 'N96' which has a mid-latitude resolution of 135 km; and the higher resolution 'N216', with the mid-latitude grid spacing of 60 km. The N96 simulations were run with 0, 1, 3, 10, 23 and 50 water tracers, whilst the N216 simulations used a smaller subset of 0, 1, 3 and 23 tracers to reduce costs. These particular numbers of water tracers were tested as 3 tracers are typically used in water isotope enabled models, 23 are used in the experiment analysed below and 50 is an arbitrary large number. As the computing time can vary due to other factors (e.g. demand on the system, network variability), each experiment was repeated three times and the mean values are used here.

175 Figure 1: Increase in wallclock time (%) due to changing the number of normal water tracers in the simulation with lower resolution (N96, black stars) and higher resolution (N216, grey crosses) experiments. The linear fit of the results is shown by the black dashed line (N96) and dotted grey line (N216).

The wallclock time increases linearly with the number of water tracers (Fig. 1), with each additional water tracer increasing the computing time by 3.1% for N96 and 3.8% for N216 on average. Therefore, the water tracers are relatively efficient, mainly as there are computationally expensive areas of the UM that they do not impact (e.g. radiation, aerosols, cloud observation simulator). The time increases are largely due to the cost of advecting the additional tracers, together with expected increases in the microphysics, convection and large-scale cloud scheme. The peak memory usage at N96 resolution increases

at 0.8% per water tracer. However, at N216, there is no significant change to the peak memory when running with 23 water tracers compared to the cost of running the model at a higher resolution overall.

#### 2.3 Prescribed-region and Scaled-flux Water Tracers

In order to test the water tracer development, various experiments are run with prescribed-region and scaled-flux water tracers. The surface evaporative flux for these water tracers ( $E_{wt}$ ) is set as

$$E_{wt}(i,j,t,i_{wt}) = \begin{cases} R_q^{sfc}(i,j,t,i_{wt}) E(i,j,t), & E(i,j,t) 

Figure 2. Mass-weighted mean of the evaporative source latitude (in degrees) of annual mean ocean-sourced precipitation evaluated from: a) the scaled-flux water tracers; b) the prescribed-region water tracers using  $10^{\circ}$  latitude bands. c) Difference (in degrees) between the scaled-flux and prescribed-region source latitude fields (a-b).

# 255

As this is a short technical test, the interpretation of the scaled-flux results is discussed later in section 3.2. Here, the focus is the differences between the two methods which are generally small; the mean absolute difference is 0.2° and the maximum absolute difference is 1.8°. This provides confidence that both types of water tracers are working correctly in the UM. It also highlights the computational efficiency of the scaled-flux method, which in this example, uses 3 tracers compared to 19 prescribed-region tracers to get comparable results. The difference plot (Fig. 2c) shows clear latitudinal bands as was also found in the equivalent ECHAM6 experiment (G24), although the UM values are smaller. The stripey difference pattern is caused by the approximation of using the mid-latitude of each band in the prescribed-region tracer calculation and reveals the improved precision of the scaled-flux approach for this calculation. The precision of the prescribed-region calculation can be improved by using more tracers with smaller latitude bands, but this makes the approach even more computationally expensive.

#### 265

260

#### 2.5 Model Experiment Setup

The water tracers are tested in an atmosphere-only historical run of the UM for the period 1979 to 2014. The 'N96' horizontal resolution of the UM is used which has 192 longitude points by 144 latitude points, with a mid-latitude resolution of 135km. There are 85 vertical levels with 50 levels below 18km and a fixed model lid at 85km above sea level. The simulation used

the AMIP sea surface temperature and sea ice concentrations (Durack and Taylor, 2017) as surface boundary conditions. The scientific configuration was GAL9.0 and the UM version was 13.3.

The experiment included 23 water tracers: 1 normal water tracer; a group of 7 prescribed-region tracers (which together cover the entire Earth's surface); and 5 groups of scaled-flux tracers (which require 3 tracers in each group). Details are given in Table 1. Source longitude is found by tracking the sine and cosine of the source longitude to avoid problems with circular data as in G24. The source longitude can then be retrieved using the arctangent of the sine field divided by the cosine field. For the purposes of this paper, the output from the two SH land water tracers (numbers 7 and 8) are combined to a single SH land source tracer.

| Water tracer | Type of water tracer | Details                                                                                |
|--------------|----------------------|----------------------------------------------------------------------------------------|
| number       |                      |                                                                                        |
| 1            | Normal water tracer  | Comparable to model prognostic water field                                             |
| 2            | Prescribed-region    | NH sea ice                                                                             |
| 3            | Prescribed-region    | SH sea ice                                                                             |
| 4            | Prescribed-region    | NH open ocean                                                                          |
| 5            | Prescribed-region    | SH open ocean                                                                          |
| 6            | Prescribed-region    | NH land                                                                                |
| 7            | Prescribed-region    | SH land (north of 60 °S)                                                               |
| 8            | Prescribed-region    | Antarctica (land south of 60 °S)                                                       |
| 9-11         | Scaled-flux          | Source latitude, with $X_{lower} = -90^{\circ}$ , $X_{upper} = 90^{\circ}$             |
| 12-14        | Scaled-flux          | sin (source longitude), with $X_{lower}$ = -1, $X_{upper}$ = 1                         |
| 15-17        | Scaled-flux          | cos (source longitude), with $X_{lower}$ = -1, $X_{upper}$ = 1                         |
| 18-20        | Scaled-flux          | Source SST, with $X_{lower}$ = -5 °C, $X_{upper}$ = 45 °C                              |
| 21.22        | Cooled flux          | Wind speed in lowest grid box, with $X_{lower} = 0$ , $X_{upper} = 32 \text{ ms}^{-1}$ |
| 21-23        | Scaled-flux          | (Not discussed here but included in the accompanying dataset)                          |

Table 1. Details of the water tracers used in the UM simulation. For the prescribed-region tracers, the third column indicates the region of evaporation that is tracked by the tracer. For the scaled-flux tracers, the third column states which evaporative property is being tracked.

280

The UM results were compared with a comparable simulation of the atmosphere model ECHAM6 (Stevens et al., 2013), forced with the same SST and sea ice fields. The ECHAM6 simulation uses a horizontal grid of 192 longitude points by 96 latitude points with 47 vertical levels reaching to 0.01 hPa (~100-110 km); therefore, it has a lower resolution latitudinally and

vertically compared with the UM simulation. The water tracer set up for ECHAM6 follows that detailed in G24 which includes the same scaled-flux tracers as in the UM simulation. As the ECHAM6 study focused on Antarctic precipitation sources, the prescribed-region fluxes are set up slightly differently to the UM study which is aiming to test the scheme globally.

The water tracers are initialised by setting the water tracer fields to the corresponding water field multiplied by  $SF(i,j,t,i_{wt})$ . Due to the relatively short residence time of water in the atmosphere (Gimeno et al., 2021), the tracers spin up quickly. For the analysis presented here, the 30-year period of 1985-2014 is analysed.

290

#### 3. Results

#### 3.1 Distribution of precipitation sources in the UM obtained from the prescribed-region water tracers

The percentage of precipitation sourced from the different evaporative surface types, as obtained from the prescribed-region tracers, is shown in Fig. 3. As expected, the majority of precipitation is sourced directly from the ocean. The amount of precipitation sourced from sea ice sublimation is small and focused over spring/summer sea ice regions. The spatial pattern of the precipitation sourced directly from land evapotranspiration (Fig. 3c and d) is comparable with similar maps from other studies: Findell et al. (2019, their figures 2b, d and f); van der Ent et al. (2014, their figure 2a); Tuinenburg et al. (2020, their figure 4); and Yoshimura et al. (2004, their figure 3 which shows a May to October mean). The precipitation over deserts is very low, but the source is still indicated here. In general, the amount of recycling over land increases downwind across continents. The largest recycling rates occur over the eastern part of Asia, with high rates also occurring in a band running northwest to southeast across central South America. Significant recycling is also diagnosed over parts of Africa and inland North America. There is a clear seasonal cycle in the land sourced precipitation (Fig. 4) with the largest values in the summer in each hemisphere, particularly in the NH, in agreement with other studies (Tuinenburg et al., 2020; Dirmeyer and Brubaker, 2007; Koster et al., 1986). This is due to evapotranspiration rates peaking in the summer months. In general, precipitation is sourced from the same hemisphere in which it falls. One exception is over the Indian monsoon region where precipitation includes a source from SH oceans, as also found in the water tracer study of Tharammal et al. (2023). Overall, the prescribed-region water tracer results look sensible.

Figure 3: Percentage of annual mean precipitation from the UM simulation (1985-2014) sourced from: a) NH open ocean evaporation (including leads); b) SH open ocean evaporation (including leads); c) NH land evapotranspiration; d) SH land evapotranspiration; e) NH sea ice sublimation; and f) SH sea ice sublimation. Black contour lines for 20, 40, 60 and 80% are shown.

Figure 4. Percentage of precipitation from the UM simulation (1985-2014) sourced from land for: a) December, January, February mean; b) March, April, May mean; c) June, July, August mean; and d) September, October, November mean. Black contour lines for 20, 40, 60 and 80% are shown.

#### 3.2 Scaled-flux water tracers comparison between UM and ECHAM6

The UM water tracer code is further evaluated by comparing scaled-flux water tracers results with those from the ECHAM6 model. Figures 5-7 compares the mean evaporative source properties of latitude, longitude and SST for *open ocean-sourced* precipitation for the two models. To aid interpretation of these figures, an example of the UM results is that the ocean-sourced precipitation falling at the grid box centred on 54.4 °S, 68.4 °W (close to the southern tip of South America) originated from a mean source latitude and longitude of 43.2 °S and 116.4 °W, with a mean source SST of 12.7 °C. The mean source latitude (Fig. 5) shows the expected pattern of precipitation being more locally sourced (in terms of latitude) in the tropics/sub-tropics and the source becoming more remote towards the poles. The dominant westerly storm tracks or easterly trade winds clearly impact on the mean source longitude at different latitudes in Fig. 6, with the mean precipitation source occurring upwind of where the precipitation falls. The mean source SST and latitude have similar spatial patterns as expected due to a strong correlation between the two fields (Fig. 7). Over the high altitudes of Antarctic and Greenland, both models show the precipitation is sourced remotely from relatively warm water compared with the source SSTs at surrounding lower altitudes. This is consistent with other findings that water vapour is sourced from relatively equatorwards/warm seas and then takes

elevated pathways to reach the high elevations of the Antarctic Plateau (e.g. Noone and Simmonds, 2002; Sodemann and Stohl, 2009; Wang et al., 2020).

The large-scale patterns of the results from the two models are reassuringly very similar. However, there are some interesting differences between the two models, which will be the subject of future work. These include the more southerly source of ocean water supplying precipitation over the Indian monsoon region and the more polewards source of precipitation over Antarctica and Greenland in the UM compared to ECHAM6 (Fig. 5). This demonstrates how water tracers can be a valuable diagnostic tool in highlighting contrasts in the hydrological cycle of different models.

340

Figure 5: Mass-weighted mean of the evaporative source latitude (in degrees) of annual mean ocean-sourced precipitation for: a) UM; and b) ECHAM6.

Figure 6: Mass-weighted mean of the evaporative source longitude (in degrees) of annual mean ocean-sourced precipitation for: a) UM; and b) ECHAM6.

Figure 7: Mass-weighted mean of the evaporative source SST (°C) of annual mean ocean-sourced precipitation for: a) UM; and b) ECHAM6.

# 3.3 UM's global hydrological cycle derived from water tracers

To illustrate how water tracers can be used to gain a simplified view of the complex GCM, Fig. 8 shows the components of the model's global hydrological cycle that can be derived from the prescribed-region tracers. The water tracers uniquely provide the amount of precipitation sourced separately from the land and the ocean (red arrows in Fig. 8). It is interesting to note that 40% of the land evaporative source precipitates out over ocean in the model. The hydrological cycle is completed in the model by the global runoff, which at steady state, should balance the net transport of atmospheric water from ocean to land (52 x 10<sup>3</sup> km<sup>3</sup> yr<sup>-1</sup>). The continental precipitation recycling ratio, as defined by van der Ent et al. (2010) as the percentage of continental precipitation that is land sourced, is 35% (as derived from the water tracer values in Fig. 8). This is slightly lower than the range of estimates from other present day studies, including 38 - 43% (Findell et al., 2019), 36% (van der Ent et al., 2014) and 51% (Tuinenburg et al., 2020).

Figure 8: Global hydrological cycle fluxes (in 10³ km³ yr¹) derived from the prescribed-region water tracer precipitation fields. The separate land and ocean sources of both the annual mean land and ocean precipitation are shown from the model simulation (1985-2014). The red arrows indicate the additional information provided by the water tracers compared to standard diagnostics. The grey dashed arrow shows the net ocean to land flux of atmospheric water. Sea ice is included in the ocean component for this diagram.

This technical paper is not intended as a scientific assessment of the UM's hydrological cycle. However, comparing Fig. 8 to observational and reanalysis based global estimates (e.g. Trenberth et al., 2007; Rodell et al., 2015; Koutsoyiannis, 2020) suggests the model's hydrological cycle is too strong with ocean precipitation and evaporation being too large. This has been stated as a common model problem and a well-known issue for the UM (Williams et al., 2017; Walters et al., 2019), and is insensitive to model resolution (Demory et al., 2014). However, Abbott et al. (2019) found a large range of global ocean precipitation ( $320 - 460 \times 10^3 \text{ km}^3 \text{ yr}^{-1}$ ) and evaporation ( $350 - 510 \times 10^3 \text{ km}^3 \text{ yr}^{-1}$ ) estimates and there are many challenges in observing these fields (Dorigo et al., 2021). Indeed, the UM values do compare well with those given in Allan et al. (2020).

The precipitation source values in Fig.8 can be compared to the model's standard net evaporative diagnostics to estimate the accuracy level of the water tracers. The net evaporation over land is  $72.29 \times 10^3 \text{ km}^3 \text{ yr}^{-1}$  compared to the water tracer land source amount of  $73.09 \times 10^3 \text{ km}^3 \text{ yr}^{-1}$ . There is a subtle difference in the definition of these two fields; the net evaporation includes the impact of negative evaporation which is not included in Fig. 8. However, preliminary investigations suggest that this is not sufficient to explain the  $0.8 \times 10^3 \text{ km}^3 \text{ yr}^{-1}$  difference. Therefore, the 1% difference can be viewed as an estimate of the accuracy of the water tracers for tracking total land evaporation. The difference between the global total water tracer

precipitation and the actual precipitation diagnostic is very small (7 x 10<sup>-8</sup> %), indicating that the water tracers are tracking all the water in the model to a high degree of accuracy. In summary, this means that an additional 0.8 x 10<sup>3</sup> km<sup>3</sup> yr<sup>-1</sup> of water has been (incorrectly) added to the land sourced water tracers during their journey from source to sink and is balanced by the same amount being removed from the other water tracers. Some inaccuracy is to be expected due to numerical errors associated with splitting the model's water into seven individual tracers, but 1% is a relatively small error.

#### 4. Conclusions and Outlook







Numerical water tracers have been successfully implemented in the UM, which track water through the following processes: surface evaporation, large-scale advection, surface exchange, boundary layer mixing, large-scale precipitation, cloud microphysics and convection. The implementation has been shown to track water to a high degree of precision, with the difference between the prognostic water and water tracer specific humidity remaining less than  $10^{-10}$  kg kg<sup>-1</sup> at the end of each timestep in a 35-year simulation. This has not been a trivial task. For example, it has required the water tracers being 'plumbed' through the entire microphysics and convection schemes to correctly track the transport and phase changes of water in these schemes. The code implementation for the normal water tracer has passed the Met Office's review and approval process and is now included in the code trunk. This helps ensure that the water tracer code will have longevity and will remain up to date with the underlying model. The new water tracer capability is relatively efficient compared to the standard run time of the UM, with one water tracer increasing the run time by 3-4%.

Tests have been carried out using prescribed-region and scaled-flux water tracers in a historical simulation and the results have been compared with the ECHAM6 model. Both types of water tracers produce sensible distributions and the scaled-flux water tracer results are comparable to ECHAM6, which provides confidence in the new UM development. There are some interesting differences in the source properties between the two models in certain regions (e.g. over South Asia, Greenland and Antarctic) and the water tracers will be used in future projects to investigate the causes of these differences. The comparison highlights that water tracers provide a valuable diagnostic tool in inter-model comparisons. It has also been demonstrated that the water tracers can be used to assess the model in an integrated and simplified manner in global hydrological cycle diagrams, including providing unique information.

The scaled-flux water tracers offer an efficient method to estimate the mean source location of the world's precipitation directly from the model. For tracing all evaporation (rather than open-ocean evaporation as done in this paper), seven tracers would be required (2 for latitude, 4 for longitude plus 1 normal water tracer), increasing the UM run time by ~22% at the N96 resolution. Therefore, including these tracers in standard climate model runs is feasible and would allow predicted changes to source locations to be easily assessed.

There are further possible uses of the new water tracer development, other than those illustrated in this work. If the water tracer precipitation is output daily, then the scaled-flux tracers can provide the mean source latitude and longitude of daily precipitation at each model grid point for each day in a model run. For example, this could be used to investigate the variability of evaporative sources for precipitation in particular regions (G24). Or this method could potentially be used to investigate sources of particular precipitation events in model simulations nudged to a reanalysis. Fiorella et al. (2021) has also shown that water tracers can be set up to track mean condensation properties or mean properties along water pathways such as the residence time or distance travelled. The water tracers could also be configured to investigate the fate of evapotranspiration from the different types of land surface included in the land model component (JULES).

In order to allow land fluxes to be investigated in more detail, work has recently been completed to add water tracers to JULES. The next major step for the water tracer development is to add isotopic fractionation processes to the UM and JULES so that water isotopes can be modelled, which is the ultimate aim for this work. Modelled water isotopes can be compared with measured values in vapour and precipitation, including those preserved in ice cores, which will provide new opportunities for model evaluation work both for present day and paleoclimates. To conclude, the new water tracer UM development is a valuable tool in understanding the hydrological cycle in past, present and future simulations.

# Appendix A




The scaled-flux water tracer method is fully derived in Fiorella et al. (2021) where the tracers are named 'Evaporative Source 440 Property Tracers'. Figure A1 is a schematic diagram to illustrate the method in a highly simplified hydrological cycle. The steps in the figure are:

- 1. The water tracer evaporation flux is set equal to the normal water evaporative flux scaled by the source property of interest, SF(i,j). Surface evaporation then adds an amount of water vapour,  $\Delta q(i,j)$ , and the scaled water tracer equivalent,  $SF(i,j)\Delta q(i,j)$ , to the atmosphere.
- 2. The water and passive water tracer are both impacted by the same advection and mixing. This means that over time, the specific humidity in a grid box has potentially several surface sources and the water tracer equivalent provides the mass weighted sum of *SF*(*i,j*) over all sources.
  - 3. Condensation processes and the subsequent precipitation do not impact the ratio of water to water tracer. The ratio is also unaffected by any re-evaporation of precipitation.
- 4. Therefore, the mass weighted mean of SF(i,j) for the precipitation falling at a particular location can be extracted from the water tracer and water precipitation values.

The hydrological cycle in the UM is obviously more complex than discussed here, with processes such as condensation happening repeatedly during a water parcel trajectory from source to sink. However, Fig. A1 still captures the water tracer behaviour during the key processes in the UM.

Figure A1: Schematic diagram to illustrate the scaled-flux water tracer method. q, qcl, E, P are specific humidity, liquid or ice condensate, surface evaporative flux and precipitation respectively. The water tracer equivalents are  $q_{wb}$ ,  $qcl_{wb}$ ,  $E_{wt}$  and  $P_{wt}$ . SF(i,j) is the scaling factor which here equals the source property that is being tracked (e.g. latitude, longitude, SST), which is scaled to be between 0 and 1 as shown in Eq. (5). All fields have multiple indices, but to reduce complexity, the only indices shown are (i,j) which indicate the surface grid box at the time of evaporation. So  $\Delta q(i,j)$  is the vapour amount originating from the evaporative flux at the surface grid box (i,j). f is the fraction of each source that contributes to the total specific humidity in a model grid box. The numbers in circles indicate various steps in the cycle which are described in the main text.

# Code and data availability

Due to intellectual property rights restrictions, we cannot provide the source code for the UM. The Met Office Unified Model is available for use under licence. A number of research organisations and national meteorological services use the UM in collaboration with the Met Office to undertake basic atmospheric process research, produce forecasts, develop the UM code,

and build and evaluate Earth system models. For further information on how to apply for a licence, see <a href="https://www.metoffice.gov.uk/research/approach/modelling-systems/unified-model">https://www.metoffice.gov.uk/research/approach/modelling-systems/unified-model</a>. The UM and/or JULES code branch(es) used in the publication have not all been submitted for review and inclusion in the UM/JULES trunk or released for general use. However, the UM/JULES code used has been made available to reviewers.

The UM water tracer precipitation data is available in the Centre for Environmental Data Analysis (CEDA) archive: https://catalogue.ceda.ac.uk/uuid/10ae416c4ccb4a90bdb5da0bbf68d4f9/

#### **Author contributions**

AM managed the water tracer code development in the UM and wrote the scientific related code. SW wrote the technical infrastructure code in the UM. AM carried out the UM run and analysis, whilst the ECHAM6 run and analysis were done by QG. GL carried out the timing tests. LS instigated and continues to manage the project to add water tracers and isotopes to the UKESM. All authors were involved in planning and discussing the code development. AM wrote the paper with contributions from all co-authors.

#### **Competing interests**

The authors declare that they have no conflict of interest.

#### 485 Acknowledgements




We thank the Met Office's Simulation Systems and Deployment Team for their support during this development and the various scientists and software scientists that reviewed the code during the Met Office's review and approval process. We acknowledge use of the Monsoon2 system, a collaborative facility supplied under the Joint Weather and Climate Research Programme, a strategic partnership between the Met Office and the Natural Environment Research Council.

# **Financial support**

The UM code development was supported by the European Union's Horizon 2020 research and innovation programme under grant agreement no. 820970 (TiPES project). Alison McLaren and Louise Sime were also supported by: NERC UK Polar Research Expertise for Science and Society: PRESCIENT (grant number NE/Y006178/1); NERC The Sensitivity of the West Antarctic Ice Sheet to +2C: SWAIS2C (grant number NE/X009386/1) and NERC SURface FluxEs In AnTarctica: SURFEIT

(grant number NE/X009319/1). Qinggang Gao was supported by the European Union's Horizon 2020 research and innovation programme under Marie Sklodowska-Curie grant agreement no. 955750 (DEEPICE project).

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
