# Peer review of "Implementation of Water Tracers in the Met Office Unified Model"

_EGUsphere, 2024_

## Referee Comment (RC1)

**Review ofMcLaren et al**

April 6, 2025

This article presents the implementation of water tacers in the Met Office Unified Model. This allows to track the origin of water vapor and precipitation and evaporative source properties. This functionality is useful for water cycle studies and is a first step towards the implementation of water isotopes, with applications for present-day water cycle and past climate. This article will serve as a reference paper for these future studies.

The paper is well-written and well-illustrated.

My comments are minor:

- l 181: I'm surprised that it takes so little memory. How can you explain this?

- l 267: latitude -> longitude?

- l 294: cite other studies that have documented this pattern using water tagging: [Koster et al., 1986, Yoshimura et al., 2004, Risi et al., 2013]

- l 299: due to larger evapotranspiration in summer?

- l 366: how can you explain this 1% difference? Especially given that you make an adjustment at the end of each time step?

**References**

[Koster et al., 1986] Koster, R., Jouzel, J., Suozzo, R., Russell, G., Broecker, W., Rind, D., and Eagleson, P. (1986). Global sources of local precipitation as determined by the NASA/GISS GCM. *Geophy. Res. Lett.*, 13 (2):121–124, DOI:10.1029/GL013i002p00121.

[Risi et al., 2013] Risi, C., Noone, D., Frankenberg, C., and Worden, J. (2013). Role of continental recycling in intraseasonal variations of continental moisture as deduced from model simulations and water vapor isotopic measurements. *Water Resour. Res.*, 49:4136–4156, doi: 10.1002/wrcr.20312.

[Yoshimura et al., 2004] Yoshimura, K., Oki, T., Ohte, N., and Kanae, S. (2004). Colored moisture analysis estimates of variations in 1998 asian monsoon water sources. *J. Meteor. Soc. Japan*, 82:1315–1329.

---

## Author Response (AR1)

**Author Response to Reviews**

We thank both reviewers for their considered reviews and constructive feedback. In summary, the main changes that we have made to the paper in response to the reviewers' comments are:

- i) An appendix containing a schematic diagram of the scaled flux water tracers has been added to help explain the method. Also, additional text giving an explicit example of the results from a single grid point have also been added to aid interpretation of figures 5-7.
- ii) The 'Conclusions' section has been changed to 'Conclusions and Outlook' with an additional paragraph detailing possible future uses of the water tracers which includes suggestions from reviewer #2.
- iii) The discussion on the water tracer error in section 3.3 has been expanded to clarify the source of the error.

Please find our responses to individual questions/comments below. Text shown in bold indicates the new text that has been added to the manuscript.

**Reviewer #1**

Line 181: I'm surprised that it takes so little memory. How can you explain this?

We are not aware of any other published results indicating the additional memory required for water tracers, so it is not possible to comment on how the UM compares with other GCMs. The only memory diagnostic that was available for our UM simulations was the *peak memory usage* of the model run, so we are unable to do any further analysis on this. The main conclusion is that adding water tracers doesn't significantly impact the *peak memory* requirement of running the model.

• Line 267: latitude -> longitude?

Yes - now corrected.

• Line 294: cite other studies that have documented this pattern using water tagging: [Koster et al., 1986; Yoshimura et al., 2004; Risi et al., 2013]

We have added the reference to Yoshimura et al. (2004) noting this is a May to Oct mean field. We have also added the reference to Koster et al. (1986) in relation to the comparison of results looking at the seasonal cycle of land sourced precipitation. Figure 1 in Risi et al. (2013) shows the percentage of *low level vapour* originating from continental evaporation, rather than precipitation. As they are not exactly the same field, we prefer not to cite Risi et al here.

• Line 299: due to larger evapotranspiration in summer?

Yes, added 'This is due to evapotranspiration rates peaking during the summer months.'

• Line 366: how can you explain this 1% difference? Especially given you make an adjustment at the end of each time step.

The adjustment made at the end of each time step, ensures that the sum of the water tracers exactly equals the amount of water in each model grid box. The 1% difference relates to tracking just the land evaporation. As the water tracer land source amount is 0.8 x 103 km3 yr-1 larger than the model's total land evaporation diagnostic, it means this net amount of water has been added (incorrectly) to the land source tracers during their journey from source to sink (and removed from the other tracers). Some inaccuracy is to be expected due to numerical errors associated with splitting the model's water into seven tracers, but we would argue that this is relatively small error.

We have added the following text to the manuscript:

'In summary, this means that an additional  $0.8 \times 10^3 \text{ km}^3 \text{ yr}^1$  of water has been (incorrectly) added to the land sourced water tracers during their journey from source to sink and is balanced by the same amount being removed from the other water tracers. Some inaccuracy is to be expected due to numerical errors associated with splitting the model's water into seven individual tracers, but 1% is a relatively small error. '

**Reviewer #2**

• General comment: I do wonder how the model performs for individual cases or tracking the sources of for example extreme precipitation events? This could be included as a possible outlook for future analysis?

We have followed your later suggestion of renaming the final section 'Conclusions and Outlook' and have added the following paragraph:

'There are further possible uses of the new water tracer development, other than those illustrated in this work. If the water tracer precipitation is output daily, then the scaled flux tracers can provide the mean source latitude and longitude of daily precipitation at each model grid point for each day in a model run. For example, this could be used to investigate the variability of evaporative sources for precipitation in particular regions (Gao et al., 2024). Or this method could potentially be used to investigate sources of particular precipitation events in model simulations nudged to a reanalysis. The water tracers could also be configured to investigate the fate of evapotranspiration from the different types of land surface included in the land model component. Fiorella et al. (2021) has also shown that water tracers can be set up to track mean condensation properties or mean properties along water pathways such as the residence time or distance travelled.'

• Last sentence abstract: this sentence is quite vague to me (especially when you haven't read the full paper), can you be more specific?

Sentence changed to:

'Finally, global model hydrological fluxes are derived from the water tracers to show the amount of precipitation sourced from the land and ocean separately, which illustrates the additional information that can be provided from the new development.'

• Line 43: There are also online models that embed numerical water tracers in regional models such as WRF-WVT (Insua-Costa & Miguez-Macho, 2018)

This is very useful to know and we have also been recently pointed to another paper using water tracer in regional models. We have added the following sentence to the introduction:

'Water tracers have also been included in high resolution regional models and used to investigate sources of specific precipitation events, such as work by Insua-Costa and Miguez-Macho (2018) and Winschall et al. (2014); the latter reference also includes a direct comparison with a Lagrangian method. '

• Line 45: One disadvantage of online water tracers if I understand correctly is that for tracking specific events you do not get the sources on a spatial grid but only per prescribed region (as the prescribed region tracer says), I think this should be mentioned here.

This is indeed true for prescribed region tracers and we have added a sentence on this:

'Traditional online water tracers that track evaporation from prescribed regions are also unable to provide detailed spatial information on individual source regions for specific events.'

However, the scaled-flux tracers can provide higher spatial information on the sources as discussed in the new paragraph in the Conclusions and Outlook.

Line 67: at --> and?

'look at' has been removed from this sentence.

• Line 146: quantify the timestep

Added '(20 mins)'.

• Line 150: the adjustment remains less than this level of 10^(-10) kg/kg at every..

Changed.

• Line 162: what do you mean with 'normal water tracer'? is that referring to prescribed water tracers?

The 'normal water tracer' is a water tracer that follows all the water in the model, i.e. it should match the model's prognostic water field. It is defined in the first paragraph of section 2.1.

• Line 190: indicate that the fields are functions

Done

• Line 200: So it is not possible to separate different land-use evapotranspiration sources? This might be a nice addition for future development

The different land-use evapotranspiration sources are not tracked in this model experiment. However, it would be relatively easy to do this and we agree it would be an interesting future development. We have added a comment on this to the Conclusions and Outlook.

• Prescribed regions; now chosen to distinguish ocean, land and ice, but this can be any region of interest? For example a country or river basin. I think it is important to make this clear in the paper that for further experiments this can be done.

Yes, the prescribed regions could be set up to trace evaporation from any geographical area. But note, this is tracking the evaporation *from* these specified regions. We have added, 'In this experiment' to the relevant sentence to stress that this is just how we set up this experiment. Further experiments are now suggested in the Conclusions and Outlook.

• Line 223: it is unclear what a 'group' exactly entails

**Added:**

'The prescribed water tracers are in a group where the prescribed regions cover the entire globe, whereas each group of scaled flux tracers contain the three tracers detailed in Eq. (5).

• Line 225-227. Twice ratio in one sentence but I think for the second 'ratio' the ratio of water tracer to water specific humidity is meant, but this is not entirely clear

Added 'water tracer to water' ratio to clarify the second ratio in the sentence.

• Line 240: A three month spin-up seems very long for introducing water tracers, is this needed and can it be clarified/quantified?

Yes, a one-month spin-up would probably be more than sufficient for the water tracers but longer is needed to ensure the atmosphere model itself has spun-up.

Figure 2: Interpretation would be easier if latitudes (and longitudes) are added to the maps.

Latitude and longitude gridlines have been added for figures 2, 5, 6 and 7 to aid the interpretation of the scaled flux water tracer results.

• Line 249: I would like some guidance (in text) on how to interpret this plot on latitudes as it is not so easy to understand, and also comes back later in the manuscript. What can we learn from the plot (besides the comparison between the different tracers)

As this is a short technical test, we prefer to delay the interpretation of the scaled flux results until the results section. We have therefore added:

'As this is a short technical test, the interpretation of the scaled flux results is discussed later in section 3.2. Here, the focus is the differences between the two methods which are generally small....'

We do appreciate that the results are difficult to interpret and we have addressed this in response to comments regarding figure 5-7 below.

• Line 269: numbers 6 and 7 --> should it be numbers 7 and 8?

Yes. Now corrected.

• Table 1: The results of water tracer number 21-23 are not shown in this manuscript right? Why include them in this table? And what is the (scientific) incentive to trace water given a wind speed limit?

You are correct that they are not used in the manuscript. However, they are in the dataset accompanying the paper so it is useful to list them here. The incentive to investigate the wind speed is due the impact of wind speed on surface evaporation. Gao et al (2024) present results from a similar tracer using the ECHAM model. We have added the following text to the table:

'(Not discussed here but included in the accompanying dataset)'

Line 284: i\_wt

Changed to iwt.

• Line 285: This argument counteracts with a spin-up time of 3 months?

As mentioned above, the spin up time used is to ensure the atmosphere model is spun-up.

• Results: My suggestion would be to re-structure the results in order of sections; starting with section 3.2 as that is mostly model intercomparison and then figure 2 and figure 5 are more close together and can be connected more easily (as they show the same variable). Then afterwards the prescribed region tracers are evaluated in terms of general hydrological cycle characteristics. First discussing the percentages of precipitation source (now section 3.1) and then the hydrological cycle fluxes (now section 3.3). I think the section headers could also need some rethinking/rephrasing (the section header Prescribed region water tracers in the UM also works for section 3.3 for example).

We appreciate this suggestion and we follow the logic of this re-arrangement. However, we would prefer not to change the order of the results for the following reason. Section 3.2 discusses precipitation sourced from the ocean only. We therefore think it is beneficial to the reader to see the results in section 3.1 beforehand, which show maps of precipitation separately sourced from the ocean and land. We believe that this helps with the interpretation of the results in section 3.2.

We have now stressed in the text relating to figure 2 that this is a short technical test of the water tracer methods and the interpretation of the scaled flux results (from a longer run) are discussed in section 3.2.

We have renamed section 3.1 as 'Distribution of precipitation sources in the UM obtained from the prescribed region water tracers'.

Figure 3: mention what the black lines in the plots represent

Added 'Black contour lines for 20, 40, 60 and 80% are shown.' Also, to the caption for figure 4.

• Line 293: add evapotranspiration to clarify à of the precipitation sourced directly from land evapotranspiration (Fig. ...)

Added.

• Figure 5: same colorscheme as figure 2 as it shows the same variable? also extend the colorbar to minimum.

Figure 2 has been changed to have the same colorscheme as figure 5. There are no values less than 60 °S, so the colorbar does not extend below the minimum labelled value.

• Figure 5 and 6 (and Figure 2); the caption says annual mean precipitation but from the text (line 314-315) I understand this should be ocean-sourced precipitation which I found confusing. Also if it is ocean-sourced precipitation why are there results over the land and ocean, then I would only expect information over the ocean? I have difficulties interpreting/understanding these figures/analyses given the current description (I know the scaled-flux is described in the methodology but still it is not fully clear to me).

We appreciate that the method and plots can be hard to interpret. We have therefore added a new appendix with a schematic diagram to illustrate the scaled flux method in a highly simplified hydrological cycle. This is intended to guide readers through the method as simply as possible. The **new appendix** is shown at the end of this document. Strictly the variables should have multiple indices (e.g. q(i,j,k,t)). However, this would make the diagram and caption very difficult to read. So for this reason, the only index that we use is ij which refers to the evaporation source grid box.

We have also added the following sentence after figures 5-7 have been introduced:

'(To aid interpretation of these figures, an example of the UM results is that the ocean-sourced precipitation falling at the grid box centred on  $54.4\,^{\circ}$ S,  $68.4\,^{\circ}$ E (close to the southern tip of South America) originated from a mean source latitude and longitude of  $43.2\,^{\circ}$ S and  $116.4\,^{\circ}$ E, with a mean source SST of  $12.7\,^{\circ}$ C.)'

We have changed the figure captions to 'Mass-weighted mean of the evaporative source latitude (in degrees) of annual mean ocean-sourced precipitation' to make this clearer. There are results over the land, as there is still ocean sourced precipitation falling over the land – as shown in the figure 3.

Line 321: source temperatures --> source sea surface temperatures

**Changed.**

• Line 322: what is meant with 'lower heights'?

Changed to 'lower altitudes'.

• Section 3.3: For me, the second Alinea on performance breaks the flow of this section and I would suggest to move the second Alinea to the end of the section and first compare the global hydrological cycle fluxes with the literature.

**Done.**

 Figure 5 - caption: 30-year simulation --> in the abstract a 35-year simulation was mentioned?

The model was run for 35 years but only the last 30 years were included in the analysis in section 3, which matched the period analysed with the ECHAM model. Caption changed to 'model simulation (1985-2014)'.

• Section 3.3: Hydrological flux results can also be compared to Demory et al. (2014)

This reference has been incorporated to the following sentence:

This has been stated as a common model problem and a well-known issue for the UM (Williams et al., 2017; Walters et al., 2019), and is insensitive to model resolution (Demory et al., 2014).

 Section 4: As this section also provides an outlook I suggest to change the name from Conclusions to Conclusions and Outlook

**Done.**

• Line 394: between the two models

**Done**

• Line 403: 22% is still quite substantial in my opinion and is not really aligned with the word 'only' in front of the percentage

Removed 'only' from sentence.

**New Appendix A**

Figure A1: Schematic diagram to illustrate the scaled flux water tracer method. q, qcl, E, P are specific humidity, liquid or ice condensate, surface evaporative flux and precipitation respectively. The water tracer equivalents are  $q_{wl}$ ,  $qcl_{wl}$ ,  $E_{wl}$  and  $P_{wl}$ . X(ij) is the source property that is being tracked (e.g. latitude, longitude, SST). The index ij indicates the surface grid box at the time of evaporation. The index ij used on other fields indicates they are evaporative fluxes or vapour amounts originating from ij. f is the fraction of each source that contributes to the total specific humidity in a model grid box. The numbers in circles indicate various steps in the cycle which are described in the main text.

The scaled flux water tracer method is fully derived in Fiorella et al. (2021) where the tracers are named 'Evaporative Source Property Tracers'. Figure A1 is a schematic diagram to illustrate the method in a highly simplified hydrological cycle. The steps in the figure are:

- 1. The water tracer evaporation is set equal to the normal water evaporative flux scaled by the source property of interest, X(ij). Surface evaporation then adds an amount of water vapour, g(ij), and the scaled water tracer equivalent, X(ij)g(ij), to the atmosphere.
- 2. The water and passive water tracer are both impacted by the same advection and mixing. This means that over time, the specific humidity in a grid box has potentially several surface sources and the water tracer equivalent field provides the mass weighted sum of X(ij) over all sources.
- 3. Condensation processes and the subsequent precipitation do not impact the ratio of water to water tracer. The ratio is also unaffected by any re-evaporation of precipitation.
- 4. Therefore, the mass weighted mean of X(ij) for the precipitation falling at a particular location can be extracted from the water tracer and water precipitation values.

The hydrological cycle in the UM is obviously more complex than discussed here, with processes such as condensation happening repeatedly during a water parcel trajectory from source to sink. However, Fig. A1 still captures the water tracer behaviour during the key processes in the UM.

**References**

Demory, Marie-Estelle, et al. "The role of horizontal resolution in simulating drivers of the global hydrological cycle." Climate dynamics 42 (2014): 2201-2225.

Insua-Costa, Damián, and Gonzalo Miguez-Macho. "A new moisture tagging capability in the Weather Research and Forecasting model: Formulation, validation and application to the 2014 Great Lake-effect snowstorm." Earth System Dynamics 9.1 (2018): 167-185.

Koster, R., Jouzel, J., Suozzo, R., Russell, G., Broecker, W., Rind, D. and Eagleson, P.: Global sources of local precipitation as determined by the NASA/GISS GCM, Geophys. Res. Lett., 13, 121–124, 1986.

Risi, C., D. Noone, C. Frankenberg, and J. Worden (2013), Role of continental recycling in intraseasonal variations of continental moisture as deduced from model simulations and water vapor isotopic measurements, Water Resour. Res., 49, 4136–4156.

Winschall, A., Pfahl, S., Sodemann, H., and Wernli, H.: Comparison of Eulerian and Lagrangian moisture source diagnostics – the flood event in eastern Europe in May 2010, Atmos. Chem. Phys., 14, 6605–6619, 2014.

Yoshimura, K., Oki, T., Ohte, N., and Kanae, S.: Colored moisture analysis estimates of variations in 1998 Asian monsoon water sources, Journal of the Meteorological Society of Japan. Ser. II, 82(5), 1315-1329, 2004.

---

## Author Response (AR2)

**Response to second review from Anonymous referee #2:**

We thank the reviewer for their further constructive feedback.

Please find below our response to the reviewer's comments. Text shown in bold indicates the new text that has been added to the manuscript.

1) I think the authors can embed the new outlook paragraph a bit better in the existing text. For example line 444. This sentence indicates future possibilities investigating the fate of evapotranspiration. This could act as a nice bridge to the next paragraph on recent developments on the land-surface model.

The new outlook paragraph has been slightly re-arranged and the connection to the next paragraph has been improved as suggested.

2) Repeating or stating the definition of ocean sourced precipitation (precipitation having its evaporative origin (source) over the ocean) would benefit the interpretation of the work. The added sentence on the interpretation of Figures 5-7 is very useful.

We have expanded the following sentence in section 2.3, to define ocean-sourced precipitation:

The scaled-flux tracers, as proposed by Fiorella et al. (2021) and illustrated in appendix A, are implemented following the approach of G24 which focusses only on precipitation that has its evaporative source over the open ocean ('ocean-sourced precipitation').

- 3) I appreciate the additional Appendix A to clarify the conceptual approach of the scaled-flux tracers. However, it also raises a few questions on how to align the notation in the Appendix with the notation in the methodology:
- a. As mentioned in the response to the review the variables should have multiple indices q(i,j,k,t), but that this is not applied in Figure A1 to make it more legible. That is fine but then I would put a remark noting this in the caption of figure A1. Further, I would also add a comma between X(i,j) in Figure A1 to avoid confusion.

The index (ij) has been changed to (i,j) and the following has been added to the caption:

All fields have multiple indices, but to reduce complexity, the only indices shown are (i,j) which indicate the surface grid box at the time of evaporation.

b. In Figure A1  $E_w$ t is determined by X(ij)E(ij) while in equation (4) in the main text  $E_w$ t is a product of the scaling factor with evaporation ( $E_w$ t =  $SF(i,j,t,i_w$ t)E(i,j,t)), which is not the same? It seems more logic to multiply evaporation with a scaling factor then a source property?

We have changed figure A1 so that X(i,j) is replaced SF(i,j) to be more consistent and added the following to the caption:

*SF(i,j)* is the scaling factor which here equals the source property that is being tracked (e.g. latitude, longitude, SST), which is scaled to be between 0 and 1 as shown in Eq. (5).

We have also added a delta sign to the surface water vapour term to indicate this is a change in water vapour caused by the evaporation.

**Updated figure A1:**

Figure A1: Schematic diagram to illustrate the scaled-flux water tracer method. q, qcl, E, P are specific humidity, liquid or ice condensate, surface evaporative flux and precipitation respectively. The water tracer equivalents are  $q_{wl}$ ,  $qcl_{wl}$ ,  $E_{wt}$  and  $P_{wt}$ . SF(i,j) is the scaling factor which here equals the source property that is being tracked (e.g. latitude, longitude, SST), which is scaled to be between 0 and 1 as shown in Eq. (5). All fields have multiple indices, but to reduce complexity, the only indices shown are (i,j) which indicate the surface grid box at the time of evaporation. So  $\Delta q(i,j)$  is the vapour amount originating from the evaporative flux at the surface grid box (i,j). f is the fraction of each source that contributes to the total specific humidity in a model grid box. The numbers in circles indicate various steps in the cycle which are described in the main text.

**Detailed comments**

1) Scaled-flux and scaled flux is used throughout in the manuscript, be consistent

'Scaled-flux' and also 'Prescribed-region' are now used consistently, when referring to the different types of water tracers. This is also consistent with Gao et al. (2024).

2) Same with ocean(-)sourced precipitation, be consistent

'Ocean-sourced precipitation' is now used consistently.

**Further changes to manuscript:**

We have changed the reference for the UM GA8.0 scientific configuration from a Met Office technical report (Xavier et al, 2024) to a preprint which has recently become available (Willett et al, 2025a).